# Nanogallium-poly(L-lactide) Composites with Contact Antibacterial Action

**DOI:** 10.3390/pharmaceutics16020228

**Published:** 2024-02-04

**Authors:** Mario Kurtjak, Marjeta Maček Kržmanc, Matjaž Spreitzer, Marija Vukomanović

**Affiliations:** Jožef Stefan Institute (JSI), Jamova cesta 39, 1000 Ljubljana, Slovenia; marjeta.macek@ijs.si (M.M.K.); matjaz.spreitzer@ijs.si (M.S.); marija.vukomanovic@ijs.si (M.V.)

**Keywords:** gallium nanoparticles, polylactide film, antibacterial composite, contact-based antimicrobial, *Pseudomonas aeruginosa*, human keratinocytes

## Abstract

In diverse biomedical and other applications of polylactide (PLA), its bacterial contamination and colonization are unwanted. For this reason, this biodegradable polymer is often combined with antibacterial agents or fillers. Here, we present a new solution of this kind. Through the process of simple solvent casting, we developed homogeneous composite films from 28 ± 5 nm oleic-acid-capped gallium nanoparticles (Ga NPs) and poly(L-lactide) and characterized their detailed morphology, crystallinity, aqueous wettability, optical and thermal properties. The addition of Ga NPs decreased the ultraviolet transparency of the films, increased their hydrophobicity, and enhanced the PLA structural ordering during solvent casting. Albeit, above the glass transition, there is an interplay of heterogeneous nucleation and retarded chain mobility through interfacial interactions. The gallium content varied from 0.08 to 2.4 weight %, and films with at least 0.8% Ga inhibited the growth of *Pseudomonas aeruginosa* PAO1 in contact, while 2.4% Ga enhanced the effect of the films to be bactericidal. This contact action was a result of unwrapping the top film layer under biological conditions and the consequent bacterial contact with the exposed Ga NPs on the surface. All the tested films showed good cytocompatibility with human HaCaT keratinocytes and enabled the adhesion and growth of these skin cells on their surfaces when coated with poly(L-lysine). These properties make the nanogallium-polyl(L-lactide) composite a promising new polymer-based material worthy of further investigation and development for biomedical and pharmaceutical applications.

## 1. Introduction

Polylactide (PLA) is a polymer integrated into diverse spheres of our life. Its excellent biocompatibility and biodegradation into non-toxic lactic acid, which is naturally present in the human body, have allowed it to be employed in many medical applications, including implantation, tissue engineering, and drug delivery [1,2,3]. However, the degradation of pure PLA inside the human body is slow (up to several years [2,4,5,6]). Therefore, it is only used internally when the strength of the material is required to last for a long time, e.g., for ligament and tendon reconstruction or orthopedic fixation devices, such as resorbable plates, pins, screws, and wires, for prolonged release of medications and in stents for vascular and urological surgery [1,2]. Nevertheless, its biodegradability is much faster than for petrochemical plastics, so PLA is becoming an increasingly popular alternative polymer in food and agricultural mulch packaging, disposable food service ware, and kitchenware due to its highly ecological production, versatile processibility, resilient aroma barrier, water-vapor barrier characteristics, and resistance to fatty foods and dairy products [2,7,8,9]. PLA is classified as generally recognized as safe (GRAS) by the United States Food and Drug Administration (FDA), is safe for all food packaging applications [2], and is also very promising when used in the form of fibers. The natural soft feel of PLA and its resistance to stains with colored beverages and sauces have also triggered its use in the textile industry outside the medical field [2,10].

Such applications are especially in full swing nowadays, when the presence and resilience of microplastics are increasingly being discovered and when the use of personal protective equipment (especially face masks) has boomed due to the COVID-19 epidemic. During the pandemic, three billion masks were discarded worldwide, putting enormous pressure on the environment [11,12]. A single mask can release 1.5 million microplastic particles [13,14,15]. Substituting the scarcely degradable plastics in the masks with PLA could solve this issue [16]. In addition, PLA also has many favorable properties for 3D printing [8,17,18], which was introduced to meet the need for the significant quantities and fast production of the protective equipment that was suddenly required [19,20].

In most applications mentioned above, the bacterial colonization of the PLA surface is a concerning issue. The hydrophobicity of PLA can enhance the adhesion of bacteria, leading to biofilms, infections, and associated complications after insertion into the body [21,22]. Moreover, bacterial colonization on food packaging can decrease the food’s shelf-life or spread foodborne diseases [23]. Additionally, bacterial contamination also occurs on the inner and outer parts of face masks [24,25]. Since PLA is sensitive to all available sterilizing methods [26] and can become easily contaminated after sterilization, antibacterial components are frequently added to it. The range of antibacterial agents introduced so far has been vast: antibiotics [27], antibacterial peptides [28,29], chitosan [23,30,31], essential oils, volatile plant extracts, and other natural antimicrobials [23,32,33,34,35], but also various inorganic nanomaterials—in particular, silver [36,37,38,39,40,41], copper [13,39], and metal oxides (TiO_2_ [39], ZnO [42,43,44], MgO [45,46], CaO [47]).

One option that has not yet been tried is gallium nanoparticles (Ga NPs). While the antimicrobial properties of ionic gallium based on its similarity to ferric ions have been known for some time, and it has been approved in certain forms for the treatment of tumorous diseases by the FDA [48], the antibacterial action of Ga NPs was demonstrated by our group only a few years ago [49]. We developed a composite of Ga NPs and hydroxyapatite, which, in direct comparison, showed better antibacterial properties against *P. aeruginosa* and lower in vitro cytotoxicity on human lung fibroblasts IMR-90 and mouse fibroblasts L929 than a morphologically similar composite of silver NPs and hydroxyapatite [49]. A comparison with the activity of the released ions showed that the inhibitory effect of the nanocomposite could not only be attributed to Ga^3+^ ions due to dissolution [49]. There are three possible sources of antibacterial activity in Ga NPs: the Ga NPs themselves, the thin gallium oxide coating, or the Ga^3+^ ions generated by the dissolution of the Ga NPs.

Compared to natural antibacterial agents, Ga NPs with virtually no volatility would have better thermal stability and a lower chance of bacterial resistance. Additionally, they could exhibit more selective activity with lower toxicity and more efficient bioelimination than other inorganic antibacterial agents. There was no significant increase in the generation of reactive oxygen species (ROS) for L929 mouse fibroblasts in the presence of the nanogallium-hydroxyapatite composite compared to hydroxyapatite alone [49]. Other studies have also shown no significant ROS generation for gallium-based liquid metal NPs [50], no in vitro cytotoxicity of Ga NPs for human cells (HeLa cells, human L-02 hepatocytes (L-02), and MDA-MB-231 cells), and no significant in vivo toxicity of Ga NPs in mice: no weight change, no deaths within 12 days, no damage to major organs and clearance from lungs, heart, and kidneys within 72 h [51,52].

In general, the combination of Ga NPs with PLA is innovative; we have not found any previous reports about such composite materials. Only the composites of Ga NPs with certain other polymers (polymethylmethacrylate [53,54], epoxy resin [55,56], polystyrene [57]) have already been investigated, but mainly for gallium phase-size correlation, phase coexistence, and undercooling; none have been investigated for their antimicrobial properties. Moreover, the methods were based on the (mostly ultrasonic) emulsification of a liquid gallium droplet, which yielded polydisperse Ga NPs. Since then, a reliable method for a controllable size of Ga NPs with a narrow size distribution has been discovered [58], which yields hydrophobic nanoparticles that are easily dispersible in chloroform, one of the best solvents for PLA. Therefore, we hypothesized that this could enable the formation of a homogeneous composite (with well-separated Ga NPs of uniform size) via simple solvent casting.

Hence, the objective of this study was to create homogeneous composites of Ga NPs and PLA and examine the influence of the nanogallium component insertion on its properties, in particular, the antibacterial activity against *P. aeruginosa*, which is one of the multidrug-resistant pathogens and the primary culprit for respiratory arrest in patients with pulmonary cysts [48].

## 2. Materials and Methods

### 2.1. Synthesis of Oleic-Acid-Capped Colloidal Gallium Nanoparticles

Oleic-acid-capped gallium nanoparticles were prepared according to Yarema et al.’s thermal decomposition procedure [58]. A total of 19 mL of dried octadec-1-ene (ODE, technical grade (90%); ACROS Organics; dried by vacuum distillation at 120 °C) was injected into a three-neck flask and held under vacuum for 1 h at 120 °C. In the meantime, 137 mg of tris(dimethylamido)gallium(III) dimer (Ga_2_(NMe_2_)_6_, 99.9% (metals basis); Alfa Aesar, Ward Hill, MA, USA) was dissolved in 3 mL of dried di-*n*-octylamine (DOA, 97%; ACROS Organics, Antwerp, Belgium), diluted with 5 mL of dried ODE and injected into a pressure-equalizing funnel connected to the flask. Then, the system was filled with nitrogen gas, and the ODE was heated to 290 °C in an inert (N_2_) atmosphere. At that moment, the Ga-precursor solution was poured immediately from the funnel into the flask, which caused a temperature drop to 240 °C and a gradual change in the reaction mixture from yellow to brown. Within 1 min, the heating mantle around the flask was substituted for an ice bath to quickly cool the brown colloidal solution below 200 °C, which terminated the further growth of the nanoparticles. Once the nanoparticle suspension reached room temperature, the system was opened, and the suspension was divided equally into two centrifuge tubes. Then, 1 mL of oleic acid (OA, technical grade (90%); Sigma Aldrich, St. Louis, MO, USA) was added into each centrifuge tube, followed by the addition of 10 mL of chloroform (for analysis; Supelco, Bellefonte, PA, USA) and 20 mL of ethanol (absolute, anhydrous, ACS, for analysis; Carlo Erba reagents, Milan, Italy). After centrifugation at 7000× *g* for 10 min, brown sediment was obtained, and the transparent supernatant was discarded. The sediment was redispersed in 1 mL of OA and 10 mL of chloroform, and the suspension was centrifuged at 8000× *g* for 15 min after the addition of 20 mL of ethanol. The supernatant (pale yellow) was discarded again, and the sediment was redispersed in OA and chloroform. After the addition of ethanol, the suspension was centrifuged at 8000× *g* for 20 min. The final sediment was resuspended in 10 mL of chloroform to give a colloidal solution that was stable for months.

### 2.2. Preparation of Ga/PLA Nanocomposites

A colloidal solution of Ga/OA nanoparticles in chloroform was mixed with a solution of poly-(L-lactide) (PLA, Resomer L207S; Evonik industries, Essen, Germany) in chloroform and poured into a glass Petri dish. Then, the solvent was slowly evaporated to yield a smooth, homogeneous disc-shaped film of a Ga/PLA nanocomposite, which was carefully removed from the Petri dish with tweezers. Different volumes of Ga/OA colloidal solution were used to prepare the Ga/PLA nanocomposites with various Ga contents: 0.08 wt.% Ga, 0.4 wt.% Ga, 0.8 wt.% Ga, 1.6 wt.% Ga and 2.4 wt.% Ga. In addition, different concentrations of PLA (2–16 g/L) were used to prepare nanocomposite films with various thicknesses. When preparing nanocomposite films of different dimensions, the total mass/surface area was kept constant. The PLA solution undergoing the same procedure but without the addition of any colloidal Ga solution was used to prepare a 0% Ga reference material (PLA film).

### 2.3. Characterization of the Nanocomposite Films

#### 2.3.1. Quantitative Determination of Gallium

The Ga content in a sample was determined using a spectrophotometric method based on the absorbance of a GaCl_4_–Rhodamine B complex, as previously described [49,59]. For the Ga content in the colloidal Ga/OA solution, a known volume of the colloid was dried in a glass vial and dissolved in concentrated (37%) HCl (for analysis; Supelco) and then diluted with deionized water (by reverse osmosis) to 6M HCl. Similarly, for confirmation of the Ga content in the composites, pieces of Ga/PLA films were weighed, and 5–8 mg was dissolved in concentrated (37%) HCl and diluted with deionized water to 6M HCl. Then, the concentration of Ga^3+^ in the dissolved samples was determined by measuring the absorbance of GaCl_4_–Rhodamine B complex in 4 mL of benzene (spectrophotometric grade (99.5+%); Alfa Aesar), which was extracted from 4 mL of diluted sample (to fit into the linear range from 0.1 to 1 mg/L Ga of the calibration curve) in 6M HCl supersaturated with NaCl and containing 0.1% Rhodamine B. Absorbance was measured using a UV-Vis-NIR Spectrophotometer Shimadzu UV-3600 from 450 to 600 nm and the maximum absorbance in this range (usually at around 564 nm) was related to Ga concentration as abs = 0.898 c(Ga) (in mg/L).

#### 2.3.2. Morphological and Spectrophotometric Analyses

The morphology of Ga/PLA films was imaged using Verios 4G HP (Thermo Fisher Scientific, Waltham, MA, USA) scanning electron microscope operating at 2 kV. The films were prepared for scanning electron microscopy (SEM) by being attached to carbon tape and coated with 8 nm of carbon. Optical properties were examined using a UV-Vis-NIR Spectrophotometer Shimadzu UV-3600 with an integrated sphere ISR-3100. Transmittance through a nanocomposite film was measured by inserting it into the beam path before entrance into the sphere. FTIR analyses were conducted with a Perkin Elmer Spectrum 400 MIR spectrophotometer (Perkin Elmer, Waltham, MA, USA) using the attenuated total-reflection (ATR) technique, with 16 scans per sample and a resolution of 4 cm^–1^. The fraction of interfacial (bound) PLA carbonyls was calculated from the IR absorbance spectra via multiple Gaussian fitting using the Origin 2015 program and by comparing the areas under peaks at around 1730 and 1750 cm^−1^ [60,61]:(1)$$\phi_{bound\;C=O}=area_{1730{cm}^{-1}}/\left(area_{1730{cm}^{-1}}+area_{1750{cm}^{-1}}\right)$$

#### 2.3.3. Crystallinity and Thermal Properties

The crystallinity of the prepared Ga/PLA films was examined using X-ray diffraction (XRD) with an Empyrean X-ray diffractometer from 10 to 40° with 400 s steps of 0.013° and Cu anode and via differential scanning calorimetry (DSC) on a Jupiter 449 simultaneous thermal analysis (STA) instrument coupled with a 403C Aëoloss mass spectrometer (MS) (Netzsch, Selb, Germany). The measurements were performed from 35 to 220 °C with a heating rate of 20 °C/min in an 80% Ar/20% O_2_ atmosphere using a TG/DSC platinum sample holder. The temperature and enthalpy calibration of the STA instrument was performed using RbNO_3_, KClO_4_, CsCl, K_2_CrO_4_, and BaCO_3_ standards. The crystallinity was then determined from the DSC curves by integrating the melting peak (using the Origin Lab 2015 software), subtracting the integral of the cold-crystallization peak, and dividing by the PLA weight fraction and melting enthalpy of 100% crystalline α-poly(L-lactide) [62], for which the value 104.5 J/g [63] was used:(2)$$X_{c,DSC}\left[\%\right]=100\cdot \frac{\Delta H_{m}-\Delta H_{cc}}{w_{PLA}\cdot \Delta H_{m}^{100\%}}$$

Crystallinity from the FTIR spectra was calculated using the following equation [64]:(3)$$X_{c,FTIR}\left[\%\right]=100\cdot area_{918-922\;cm^{-1}}/\left(area_{918-922\;cm^{-1}}+area_{956\;cm^{-1}}\right)$$

Areas were determined by integration of peaks (using the Origin Lab 2015 software) after conversion of the recorded FTIR spectra to absorbance and manual baseline determination.

#### 2.3.4. Surface Wettability and Mechanical Properties

Surface wettability of PLA and Ga/PLA films was measured on a contact angle meter Theta Lite (Biolin Scientific, Gothenburg, Sweden) by putting water droplets of 5 µL onto the surface of a film, taking photos of the droplets and fitting the images (using the Attension theta 4.1.0 software) to obtain their contact angles. The arithmetic mean of the left and right contact angles was considered, and at least 8 replicate water droplets were put on both sides of the film to obtain the average contact angle and its standard deviation.

Young’s moduli of the films were calculated from the slopes of the stress–strain curves in the linear response range, which were measured in a Mettler Toledo (Columbus, OH, USA) DMA/SDTA861e dynamic mechanical analyzer (DMA) under 1 Hz frequency and 1–100 µm displacement range.

### 2.4. Release of Ga from the Nanocomposites in Conditions of the Antibacterial Tests

Ga release from the nanocomposites under the conditions of a planktonic growth monitoring antibacterial test (Section 2.5.2.) was conducted in an H1 Hybrid Multi-mode Microplate Reader (Synergy; Biotek, Winooski, VT, USA) at 237 rpm orbital shaking. After 24 h in Mueller–Hinton (MH) medium at 37 °C, the films were removed, transferred to deionized water, wiped on a paper towel, and dried in air. Then, they were sputtered with carbon and visualized by SEM. Meanwhile, the remaining liquids in the 96-well microplate were analyzed for Ga content after the film removal. Ga release under the conditions of the contact antibacterial test was conducted by placing a 5 µL MH droplet onto the surface of a film inside a Petri dish, incubating in a humid atmosphere at 37 °C for 24 h, as described in Section 2.5.3, and withdrawing the droplet (with a volume of 2–5 µL) for analysis of the Ga content.

The concentration of Ga^3+^ in the liquids of the ion-release study was determined by first mixing equal volumes (100–200 µL and 2–5 µL, respectively) of the liquid and concentrated (37%) HCl, supersaturating with NaCl and then diluting with 6 M HCl and the Rhodamine B (powder; Alfa Aesar) solution to a volume of 0.8 mL and extracting to 0.8 mL of benzene. The measured absorbance of the GaCl_4_-rhodamine B complex was converted to Ga concentration, as explained in Section 2.3.1.

### 2.5. Antibacterial Tests

#### 2.5.1. Agar Diffusion Test

An overnight culture of *P. aeruginosa* PAO1 (wild type, for details, see [65]) was grown in 2 mL of Luria–Bertani (LB, Miller; Carl Roth, Karlsruhe, Germany) medium at 37 °C and 250 rpm orbital shaking in a MaxQ 4000 shaking incubator (Thermo Scientific, Waltham, MA, USA) until the end of the logarithmic phase. Then, it was centrifuged at 10,000× *g*, the pellet was resuspended in 2 mL of phosphate-buffered saline (PBS, tablet; Sigma-Aldrich, St. Louis, MA, USA) and diluted with PBS to match the turbidity (optical density (OD)/absorbance at 600 nm) of the 0.5 McFarland standard [66], which was prepared from BaCl_2_ and H_2_SO_4_ and measured to have absorbance (OD600) of 0.063 ± 0.003. Next, 100 µL of the diluted culture was spread onto an MH agar plate with a glass Drigalski spatula to obtain a semi-confluent growth of bacteria. Then, three pieces of Ga/PLA film were put onto the MH agar plate with bacteria. The plate with the inoculum and the films was incubated for 24 h at 37 °C in an incubator (Kambič, Semič, Slovenia). Any area around the samples devoid of bacterial colonies (inhibition zone) would indicate the antimicrobial properties of these samples.

#### 2.5.2. Inhibition of Planktonic Bacterial Growth

An overnight culture of PAO1 was grown in LB medium at 37 °C and 250 rpm orbital shaking in a MaxQ 4000 shaking incubator (Thermo Scientific) until the end of the logarithmic phase. Then, it was diluted with MH broth (for microbiology; Sigma-Aldrich) to a concentration of around 5 × 10^5^ cfu/mL (assuming that 10^8^ cfu/mL gives OD600 ≈ 0.1). The 5×5 mm pieces were cut from a sample film (PLA or Ga/PLA) with a scalpel aseptically and inserted with sterile tweezers into the diluted bacterial liquid (200 µL for each film piece) in a 96-well flat-bottom microtiter plate. The plate was incubated at 37 °C and 237 rpm orbital shaking in a Synergy H1 Hybrid Multi-mode Microplate Reader (Biotek, Winooski, VT, USA), and planktonic bacterial growth was monitored by measuring OD600 every hour. The OD600 values of samples in a sterile MH medium were subtracted from the values of samples with bacteria.

#### 2.5.3. Contact Antibacterial Action

A contact antibacterial test was conducted according to the ISO 22196/JIS Z 2801 standard [67] but on smaller samples and a lower volume of bacteria. Nevertheless, the condition of 6.2 × 10^3^–2.5 × 10^4^ cfu/cm^2^ was met and the criterion for validity of the test (average number of immediately counted bacteria (t = 0 values) between 6.2 × 10^3^ and 2.5 × 10^4^ cfu/cm^2^, as well as the logarithmic difference between the maximal and minimal number of counted bacteria (on the sample) not being bigger than 0.2 logarithmic values of the average number of bacteria) were fulfilled. Moreover, the PLA was a very good reference material according to the standard recommendations.

An overnight culture of PAO1 was grown in LB medium at 37 °C and 250 rpm orbital shaking in a MaxQ 4000 shaking incubator (Thermo Scientific) until the end of the logarithmic phase. Then, it was diluted with MH broth (for microbiology; Sigma-Aldrich) to a concentration of around 5 × 10^5^ cfu/mL (assuming that 10^8^ cfu/mL gives OD600 ≈ 0.1). Eight 5×5 mm pieces were cut from a sample film with a scalpel aseptically, and 5 µL of bacterial suspension (5 × 10^5^ cfu/mL) was dropped onto each piece. Thin films were wrapped around the droplet, while thicker films were covered with a piece of sterile polyethylene film to spread the bacteria and keep them in good contact with the tested films. Four films were immediately transferred together with the droplet into 495 µL of PBS, and the bacteria were washed/detached from the film into the PBS solution by vortexing for 30 s and then 10 s of ultrasonication on an ultrasonic bath. This 100× diluted bacterial suspension was further diluted 10× and 100×, after which six 10 µL droplets from each dilution were dropped onto an LB agar plate, and colonies grown from each droplet after 24 h of incubation at 37 °C were counted. The remaining four pieces of each sample film (PLA or Ga/PLA with various Ga contents) were incubated for 24 h at 37 °C inside a plastic box with a wet paper towel on the internal walls to keep high humidity and prevent the bacterial droplets on the films from drying. After 24 h, the films with droplets were transferred into 495 µL of PBS and further underwent the same bacterial removal, dilution, and colony counting procedure as the four films at time zero (t = 0). The counted colonies were converted to cfu/mm^2^, and the average values with a standard deviation of four replicates at t = 0 and t = 24 h were provided as the final results for each sample film. Antibacterial activity was also expressed as the difference between the logarithmic change in bacterial number within 24 h on the samples and such change on the PLA reference film.

#### 2.5.4. Contact Bactericidal Action

Contact bactericidal action under conditions without nutrients for bacterial growth was explored in a similar way as contact antibacterial action in a nutrient medium (Section 2.5.3.), but it was performed in PBS instead of MH, and a higher bacterial concentration (around 10^8^ cfu/mL) was used. A total of 2 mL of overnight PAO1 culture was centrifuged for 10 min at 10,000× *g,* and the pellet was resuspended in 2 mL of sterile PBS. Then, it was diluted with PBS to OD600 ≈ 0.1. Films of PLA or Ga/PLA were inoculated with this bacterial suspension and further treated as described in Section 2.5.3. However, due to higher concentrations of PAO1, 10×, 100×, 1000×, 10,000×, and 100,000× dilutions were applied onto agar plates for colony counting.

Additionally, bacterial survivability after 24 h was evaluated by their live/dead fluorescent staining using a Live/Dead^®^ BacLight^TM^ Bacterial Viability Kit L7007 “for microscopy” (Molecular Probes, Eugene, OR, USA). Then, a 2× concentrated (with regards to the procedure) mixture of both dyes (1 µL of syto9 and 1 µL of propidium iodide into 331 µL of sterile PBS) was prepared, and 5 µL of this mixture was mixed with the 5 µL bacteria-containing droplet on a sample film. After 15 min of incubation at room temperature in a dark and wet place, 5 µL was removed from the film, and the film with stained bacteria on it was flipped over onto the microscope glass and covered. Then, fluorescent bacteria on the films were observed in a Nikon Eclipse Ti-U inverted microscope (Nikon, Tokio, Japan). Samples for microscopy imaging of bacteria were not covered with polyethylene during the 24 h incubation; the bacterial droplet was smeared over the films. A comparison between the survivability of the covered and smeared bacteria did not reveal any significant differences in the number of bacteria for the same material (see Appendix A).

#### 2.5.5. Scanning Electron Microscopy of Bacteria on the Films

Films with bacteria from both contact tests (after 24 h in the test from Section 2.5.3. and 0 h and 24 h in the test from Section 2.5.4.) were transferred consecutively into 2.5% glutaraldehyde in PBS (for 30 min), sterile 0.9% NaCl in water (15 min), sterile 0.9% NaCl in water (15 min), 30% ethanol in sterile water (15 min), 50% ethanol (15 min), 70% ethanol (15 min), 90% ethanol (15 min), 100% ethanol (1 h), and liquid CO_2_. The last solvent was removed as supercritical fluid in a critical point drier (K850; Quorum Technologies, Sacramento, CA, USA).

### 2.6. Cytotoxicity Tests

HaCaT human keratinocyte cells (ATCC PCS-200-011) were confluently grown from a frozen low passage in a tissue-treated T25 flask containing 10 mL of Dulbecco’s Modified Eagle’s Medium (DMEM, high glucose, sterile-filtered, suitable for cell culture; Sigma Aldrich) supplemented with 10% Fetal Bovine Serum (FBS, heat-inactivated, origin: Brazil, Gibco, Paisley, UK) and 1% Pen Strep (Penicillin Streptomycin; Gibco). Then, they were washed three times with Dulbecco’s Phosphate-Buffered Saline (DPBS, with MgCl_2_ and CaCl_2_; Sigma Aldrich), detached by trypsinization with TrypLE Select (1X, Gibco, Grand Island, NY, USA), centrifuged at 200× *g*, resuspended in full DMEM (DMEM + 10% FBS + 1% Pen Strep), and diluted 5 times in full DMEM and seeded in a tissue-treated 6-well plate. Confluently grown cells on the bottom of the wells were washed, detached, and resuspended in the same way as described above and diluted 8 times during replating for each cytotoxicity test.

#### 2.6.1. Growth of HaCaT Cells in the Presence of Films

The diluted HaCaT cells were seeded (20,000 cells/well) onto a tissue-culture-treated 24-well plate and grown in a humidified 5% CO_2_ atmosphere at 37 °C (in MCO-19AIC(UV)-PE incubator; Panasonic, Oizumi, Japan) until approximately 80% confluency. Then, 10 vol.% of Presto Blue Cell Viability Reagent was added into each well to measure cellular growth before treatment (based on fluorescence at 590 nm with excitation at 560 nm after 1 h of incubation). The dye was washed away with DPBS (2×), fresh full DMEM was added, and a 1 × 1 cm piece of PLA or Ga/PLA film was inserted. Then, the incubation of cells continued for an additional 24 h in the presence of the films, after which Presto Blue Cell Viability Reagent was added, and fluorescence was measured again in the same way after 1 h of incubation to determine the cellular growth after treatment with the films. The test was performed in three parallels. References were cells grown without films and full DMEM incubated without cells.

#### 2.6.2. Growth of HaCaT Cells on the Films

Before testing, 1 × 1 cm pieces of PLA or Ga/PLA films were immersed in 300 μL of poly(L-lysine) solution (molecular weight 70,000–150,000, 0.01%, sterile-filtered; Bio Reagent, Sigma, UK) overnight and washed with full DMEM. The film squares were seeded with HaCaT cells (diluted to 1:8 in full DMEM) inside 24-well plates and incubated at 37 °C and 5% CO_2_ for 24 h in MCO-19AIC(UV)-PE incubator (Panasonic). Then, the films with cells on the surface were moved into a new 24-well plate, and cell attachment was measured using Presto Blue Cell Viability Reagent (according to the provided protocol, with 10 wt.% reagent concentration, 1 h incubation, and a fluorescence measurement of emission at 590 nm after excitation at 560 nm). After detecting viable cells adhered on the films, they were washed with DPBS, supplied with fresh full DMEM, and incubated in a CO_2_ incubator for an additional 24 h, after which the protocol of cellular viability measurement was repeated. This analysis of cellular growth continued for 3 days and was performed on three parallels of the same film.

#### 2.6.3. Scanning Electron Microscopy of HaCaT Cells on the Films

After 3 days of HaCaT cellular growth on PLA and Ga/PLA films, the DMEM was removed and substituted by 0.6 mL of 2.5% glutaraldehyde in DPBS. After two hours of soaking in the fixative at room temperature, HaCaT cells were transferred together with the films into 0.6 mL of DPBS, which was exchanged with 0.5 mL of fresh DPBS after 5 min and once again after 5 min. Then, gradual dehydration followed, with the sequential introduction and removal of the following liquids (in 0.5 mL amounts) [68]: 10%, 20%, 30%, 40%, 50%, 60%, 70%, 70%, 80%, 100%, 100% ethanol, 1:1 (1,1,1,3,3,3-Hexamethyldisilazane (HMDS)):ethanol mixture and 100% HMDS. The samples were air-dried from HMDS, glued onto carbon tape, and coated with 6 nm of Pt before observation under SEM.

## 3. Results

### 3.1. Physico-Chemical Properties of Ga/PLA Nanocomposites

#### 3.1.1. Appearance, Composition, and Optical Properties

The Ga/PLA nanocomposites were prepared as disc-shaped films (inset in Figure 1a) by solvent casting in a glass Petri dish. The dimensions of the discs could be modified with the dimensions of the Petri dish, while the thickness of the film was affected by the concentration of PLA. The dimensions measured were: 9 µg/mm^2^ yielded films with around 15 µm average thickness and 18 µg/mm^2^ yielded films with around 30 µm average thickness. Nanogallium content in the composites varied from 0.08 to 2.4 wt.% Ga through changing the volume of the colloidal solution of Ga/OA NPs (with the size of 28 ± 5 nm) that was added to the solution of PLA. Ga/OA NPs had an almost normal narrow size distribution with an 18% coefficient of variation (Appendix A), and OA bonding to their surface was confirmed using FTIR (Appendix A). The addition of nanogallium gave the films a characteristic beige/gold color, which turned browner with the increasing fraction of Ga (Figure 1a). This is a consequence of gallium surface plasmon resonance (SPR), which is in the UV range and thus decreases the UV transparency of the films (Figure 1a). The SPR peak is clearly distinct in the transmittance spectra of thinner films (Figure 1a), and its intensity increases with increasing Ga content, which is particularly evident in the absorbance spectra after subtracting the absorbance of PLA (Appendix A). It indicates the good separation of Ga NPs within the composite and a low degree of agglomeration. The position of the Ga SPR peak at around 255 nm with respect to simulated SPR for 30 nm Ga NPs in the PLA matrix (at 239 nm; the calculated UV/VIS spectrum is added in Appendix A) is red shifted due to the thin (~2 nm) gallium oxide/hydroxide shell on the surface (Appendix A) and it matches well with the previously reported SPR of similarly prepared Ga/OA NPs [58].

The ATR FTIR spectra of PLA and Ga/PLA (Figure 1b and Appendix A) reveal characteristic vibrational bands for PLA [69] at around 3000 (*ν*_as_ (CH_3_)), 2950 (*ν*_s_ (CH_3_)), 2880 (*ν* (CH)—weak, at 2850 for Ga/PLA), 1752 (*ν* (C=O)), 1450 (*δ*_as_ (CH_3_)), 1385 (*δ*_s_ (CH_3_)), 1360 (*δ*_1_ (CH)), 1300 (*δ*_2_ (CH)—weak), 1270 (*δ* (CH)+ *ν* (C-O-C)), 1210 + 1180 (*ν*_as_ (C-O-C), doublet), 1130 (*ρ*_as_ (CH_3_)), 1080 (*ν*_s_ (C-O-C)), 1044 (*ν* (C-CH_3_), 960 + 920 (*ρ* (CH_3_)+ *ν* (CC)), 870 (*ν* (C-COO)), 760 (*δ* (C=O)) and 705 cm^−1^ (*γ*(C=O)). However, an additional peak at around 1560 cm^−1^ is evident especially for Ga/PLA with the highest Ga content, which can be attributed to the OA bidentate connection to the surface of Ga NPs (see also Appendix A) [70,71,72]. We attempted to estimate the relative interaction of PLA with Ga/OA NPs in the nanocomposites through C=O groups by deconvolution of the *ν* (C=O) vibration band peak into two Gaussian peaks—one at around 1725 cm^−1^ (arising from the bound PLA carbonyls in a material) and the central, most prominent peak at around 1750 cm^−1^—and calculating the ratio of their areas [61,73,74]. However, an additional Gaussian peak at a higher wavenumber had to be introduced for a successful fit. In fact, the shoulder on the higher frequency side of the peak originating from various amorphous and crystalline PLA phases [75,76,77] was even more pronounced than the broadening towards lower wavenumbers. This interfered with the analysis of bound carbonyl fraction as the higher-wavenumber peak affected the 1725/1750 area ratio and also the positions of the peaks, so we obtained a very large discrepancy and very high values for several PLA samples. Therefore, we fixed the two additional peaks at 1725 and 1775 cm^−1^, respectively, and varied only their height and width during fitting. This decreased the error and values for the pure PLA. However, we still noticed that the bound carbonyl fraction was somewhat connected with the crystallinity of the measured part of the film (the more crystalline part of the same sample often yielded lower values than the less crystalline one), especially in the case of Ga/PLA. Nevertheless, an increase in this value up to 0.8 wt.% Ga was generally observed, and then there was a decrease with a further increase in Ga content, indicating that interfacial interaction through C=O groups of PLA occurred most frequently in Ga/PLA0.8%. $\phi_{bound\;C=O}$ values from the spectra of the films in Figure 1b and Appendix A are 0.11 for PLA, 0.11 for Ga/PLA0.08%, 0.20 for Ga/PLA0.4%, 0.23 for Ga/PLA0.8%, 0.20 for Ga/PLA1.6%, and 0.15 for Ga/PLA2.4%, whereas on average we obtained 0.14 ± 0.06 for PLA, 0.15 ± 0.05 for Ga/PLA0.08%, 0.14 ± 0.04 for Ga/PLA0.4%, 0.21 ± 0.08 for Ga/PLA0.8%, 0.22 ± 0.04 for Ga/PLA1.6%, and 0.15 ± 0.04 for Ga/PLA2.4%.

#### 3.1.2. Detailed Morphology

The detailed investigation of Ga/PLA0.8% morphology via scanning electron microscopy revealed embedded bright Ga nanospheres in the darker wrinkled polymer, as individual separated NPs, pairs of NPs, and clusters of up to seven Ga NPs, but never as agglomerates one on top of the other, always in the same plane and separated without overlapping (Figure 2a). The degree of agglomeration increased with the increasing Ga content. The dispersion of Ga NPs within the composite was even more evident in backscattered electron images (Figure 2b), from which their size distribution was calculated. A comparison with the initial Ga/OA NPs confirmed that there was no change in the size distribution of Ga NPs during the formation of Ga/PLA composites. A slight increase from 28 ± 5 nm in the case of colloidal Ga/OA NPs (Appendix A) to 30 ± 5 nm Ga NPs in the composite (Figure 2c) is within the error of the size determination method. The EDXS analysis proved that the bright spheres were made of gallium, and their surroundings did not contain any gallium (Figure 2d).

#### 3.1.3. Crystallinity and Thermal Properties

Polymer crystallinity in the films was explored using FTIR, XRD, and DSC (Figure 3 and Table 1). No crystalline phases were detected in X-ray diffractograms of PLA and Ga/PLA0.4%, but an amorphous hump in the range up to around 25° 2*θ* was raised in the latter (Figure 3a). This is at a lower angle than the previously observed hump in the XRD of liquid Ga/OA NPs [58], which would appear above 30° 2*θ* for the Cu anode. Therefore, it cannot be attributed to the nanogallium, but rather to the polymer phase. Following the increase in the Ga content to 0.8 wt.%, two distinct diffraction maxima appeared, the highest at 16.6° and a lower one at 19.0° 2*θ* (Figure 3a). However, additional lower maxima at around 13, 15, 22, and 25° 2*θ* could also be discerned. This indicates the existence of the α-poly(L-lactide) crystalline phase in this nanocomposite [78], but an additional α’ phase cannot be excluded since the most intense, (200)/(110), and the second most intense, (203), diffraction maxima are very wide, and the diffraction close to 25° is too weak and too noisy [79] with a further increase in Ga content to 2.4 wt.%, the crystallinity is no longer enhanced–the background is raised, and only (200)/(110) diffraction can be clearly distinguished.

We also evaluated the crystallinity from the FTIR spectra (based on the ratio of 918–921 and 956 cm^−1^ peak areas, see Table 1) and by DSC, which provided some additional information about the amorphous phase and thermal properties of the composites (Figure 3b). DSC curves of all films exhibited typical thermal physical processes for poly(L-lactide) [2,62,80,81]: glass transition between 65 and 75 °C with additional endothermic enthalpy relaxation in the same range, exothermic cold crystallization between 90 and 120 °C, exothermic α’- α transformation between 155 and 170 °C and melting at around 180 °C. The most evident effect of Ga/OK NPs in Figure 3b is the decrease in the temperature of cold crystallization (*T*_cc_) with the increasing Ga content. There is also a notable difference in the melting temperature (*T*_m_), which is slightly higher for PLA and Ga/PLA0.8%. This can be related to the higher crystallinity of Ga/PLA0.8% and the higher content of α-PLA in it on the one hand (Figure 3a) and the higher *T*_cc_ of PLA on the other hand, leading to more perfect crystals and more α-PLA, which has a higher melting point than the α’ phase [82]. PLA is known for predominantly crystallizing into α’ at temperatures below 100 °C, while α-PLA is the prevailing phase above 105 °C [83]. This is confirmed by comparing the enthalpies of α’-α transformation (Δ*H*_α’-α_) in Table 1, which are evidently lower for PLA and Ga/PLA0.8% compared to Ga/PLA0.4% and Ga/PLA2.4%. The glass transition (*T*_g_ and Δ*C_p_* at *T*_g_) was difficult to analyze due to extra enthalpy relaxation in the same temperature range. This might be the reason for unusually high Δ*C_p_* values (almost two times higher for PLA than the commonly referred value for 100% amorphous PLA [40,73]). On the other hand, even higher values (up to 1.6 J/(gK)) have been reported [2,62]. Nevertheless, Ga/PLA films generally exhibited smaller Δ*C_p_* at *T*_g_ (Table 1), the minimum for Ga/PLA0.4%, while *T*_g_ was more or less the same for all materials (it was higher by 1 °C for Ga/PLA0.4% and Ga/PLA2.4% and lower by 1 °C for Ga/PLA0.8%). The crystallinity fraction $X_{c,DSC}$ calculated with Equation (2) agreed well with the crystallinity $X_{c,FTIR}$ determined from FTIR spectra using Equation (3). Both increase with the increasing Ga content up to 0.8 wt.% and then decrease with a further increase to 2.4 wt.% (Table 1), which is in agreement with XRD (Figure 3a). Enthalpy of cold crystallization Δ*H*_cc_ shows the opposite trend—the minimum at 0.8 wt.%, while *T*_α’-α_ follows the trend of *T*_cc_.

#### 3.1.4. Surface Aqueous Wettability and Young’s Modulus

Table 1 also contains the contact angle with water (*θ*_H2O_) and Young’s modulus (E) for PLA and Ga/PLA nanocomposite films with varying Ga content. There is a correlation between the hydrophobicity of the surface and the nanogallium fraction in the film. The contact angle increased from around 70° for pure PLA to around 90° for Ga/PLA2.4%. Young’s modulus increased with the introduction of nanogallium (higher E for all Ga/PLA in comparison with PLA), but instead of a monotonous trend with increasing Ga content, there was a maximum at 0.4 wt.%.

### 3.2. Antibacterial Properties of Ga/PLA Nanocomposites against Pseudomonas aeruginosa

#### 3.2.1. Inhibition of Planktonic and Colonial Growth

First, we examined if there was any remote influence of Ga/PLA films on bacteria, i.e., on their colonial or planktonic growth, which could potentially be caused by the diffusion of released Ga NPs or Ga^3+^ ions into solid or liquid media. As is evident in Figure 4a, pieces of Ga/PLA1.6% caused no zone of inhibition in the semiconfluent *P. aeruginosa* bacteria on agar. Similarly, PAO1 bacteria grew normally in a growth medium in the presence of Ga/PLA films; there were no signs of inhibition (Figure 4b).

#### 3.2.2. Contact Antibacterial Action in a Nutrient Medium

However, Ga/PLA nanocomposites impacted the bacteria in direct contact (Figure 5). While such bacteria could grow normally on PLA (as a reference film), their growth was fully inhibited on Ga/PLA with 0.8 wt.% Ga. Moreover, Ga/PLA1.6% and Ga/PLA2.4% even exhibited a bactericidal effect, with Ga/PLA2.4% depleting virtually all the bacteria that were put on it in this experiment (Figure 5a). Expressed in antibacterial activity values: around 0 for 0.08%, 0.1 for 0.4%, 4.6 for 0.8%, 5.5 for 1.6%, and 6 for 2.4 wt.% Ga. A comparison of two films with equal Ga content and different thicknesses (Figure 5b) revealed that the observed antibacterial effect of Ga/PLA was a consequence of the surface composition rather than the overall Ga mass. After the antibacterial test, a detailed inspection of the films discovered large connected biofilm-like clusters of bacteria attached to the surface of PLA (Figure 5c) and only what appeared to be the remnants of destroyed bacteria on the surface of Ga/PLA1.6% (Figure 5d).

#### 3.2.3. Contact Bactericidal Action of Ga/PLA2.4% in PBS

Furthermore, we examined the contact bactericidal activity of the antibacterial Ga/PLA films on a larger number of bacteria (around 200× larger) and in a PBS medium without the necessary nutrients for bacteria to grow (Figure 6). Their number remained virtually the same on PLA, Ga/PLA0.8%, and Ga/PLA1.6%, while the Ga/PLA2.4% nanocomposite reduced it around 100 times within 24 h (Figure 6a). Live/dead staining of bacteria on the surface of this nanocomposite confirmed a fraction of dead bacteria, which was not detected on PLA (Figure 6b).

Therefore, we investigated how this killing effect of Ga/PLA2.4% was expressed in the morphological changes in bacteria (Figure 7). The PAO1 bacteria appeared intact and with very similar outer morphology on the PLA and Ga/PLA2.4% when they were not subjected to 24 h incubation (Figure 7a,c). However, notable morphological differences emerged during 24 h of contact with the films at 37 °C. On the PLA (Figure 7b), bacteria slightly shrank but retained their integrity, and they established connections through 1–2 µm long intercellular nanotubes with diameters around 30–40 nm, which is a sign of cross-feeding during starvation [84]. By contrast, they became mostly severely damaged on Ga/PLA2.4% after 24 h, and they could not help themselves through intercellular connections (Figure 7d).

### 3.3. Cytocompatibility of Ga/PLA Nanocomposites with HaCaT Keratinocytes

After revealing their contact antibacterial properties, we tested the cytocompatibility of Ga/PLA nanocomposites with human HaCaT keratinocytes. First, we grew the cells on a standard adhesion-promoting surface in tissue-culture-treated 6-well plates and checked their 24 h survivability in the presence of PLA or Ga/PLA films with different Ga contents (Figure 8a). As we can see, the HaCaT cells remained normally metabolically active in the presence of PLA or any of the tested Ga/PLA nanocomposites. There were no statistically significant differences between the two films.

Thus, we tried to grow the cells directly on the films. However, this was not possible even on solvent-cast PLA film without its prior coating with poly(L-lysine), which reduced its hydrophobicity (see Appendix A) and made its surface attractive for keratinocyte adhesion [85]. We then monitored the growth of the cells on PLA and Ga-PLA films for 3 days by measuring their metabolic activity (Figure 8b).

There was an obvious increase in the number of cells on all tested films during the three-day period. However, after the first day, the metabolic activity remained virtually the same on Ga/PLA2.4%, although not significantly different from PLA (*p* = 0.08). However, the keratinocytes then caught up during the second day, after which Ga/PLA0.08% exhibited the lowest metabolic activity. After three days, the Ga/PLA2.4% value was again the lowest and was almost statistically significantly different from PLA (*p* = 0.055). The average metabolic activity was always the largest on the pure PLA but was also not significantly lower on the composites, which indicates that the cells could attach and divide on the surface of all the Ga/PLA nanocomposites. This was confirmed by examining their morphology under optical (Appendix A) and scanning electron microscopes (Figure 9), which both revealed a good confluent spreading of HaCaT cells on the surface and forming a tissue-like monolayer. All cells exhibited many microvilli and filopodia on the surface. They appeared flatter (less convex) and smoother only on Ga/PLA2.4%—with fewer microvilli and more fragments on top (which resembled the remains of dead/damaged cells), but still well connected with many filopodia, which also extended to the Ga/PLA surface (Figure 9d).

### 3.4. Ga Release from Ga/PLA Nanocomposites in Bacterial Growth Medium

To better understand the observed contact antibacterial activity, we also investigated the released Ga under the conditions of antibacterial tests. The concentration of released Ga from a film soaking in MH broth for 24 h at 37 °C and orbital shaking increased almost linearly with the increasing Ga content (Figure 10a). However, it was well below 15 µM even for Ga/PLA1.6%, which would still be more than 100 times lower than the reported MIC for Ga NPs and ionic Ga against this *P. aeruginosa* strain [49]. Thus, it is unsurprising that there was no inhibition of PAO1 in the planktonic test (Figure 4b). When converted to a fraction of the initial Ga content (based on dimensions and mass/area), around 6–12% of the contained Ga was released into the MH medium. On the other hand, measurement of the released Ga in the droplet under conditions of the contact antibacterial test revealed evidently higher concentrations: 0.12 ± 0.01 mM Ga from Ga/PLA0.8% and 0.8 ± 0.1 mM Ga from Ga/PLA2.4%, although the fraction of released Ga was actually around two times lower than when the film was floating in the liquid medium. The examination of Ga/PLA0.8% morphology after exposure to soaking and shaking in MH broth at 37 °C was also quite informative. SEM images revealed many empty holes in the shapes and sizes of Ga/OA NPs and their agglomerates (Figure 10b), which obviously emerged after Ga/OA NPs (perhaps even Ga/OA/PLA NPs) were released from the nanocomposite into the MH medium. A fraction of the Ga/OA NPs remained inside, under a layer of PLA, while coated nanoparticles (possibly Ga/OA/PLA NPs) on top of the nanocomposite film could also be found, especially on unwashed films and films subjected to static contact test conditions (Appendix A).

## 4. Discussion

Starting from stable colloidal solutions of Ga/OA NPs and PLA in chloroform, we prepared homogeneous Ga/PLA composites. They were stiffer and more hydrophobic than PLA, and the SPR of the Ga NPs rendered the films less UV-transparent, which would be a favorable property for food packaging applications.

At first glance, it appears obvious that the introduction of hydrophobic OA-capped nanoparticles would decrease the aqueous wettability. However, Ga/OA NPs in the nanocomposite are generally coated with PLA, which then is the top surface material in contact with water. With the OA polar part attached to the Ga/Ga-oxide surface (via strong bidentate bonding to Ga, indicated by vertical stripes, or via hydrogen bonding to O, marked with dashed lines in Figure 11a), van der Waals interactions of OA with PLA (dotted lines in Figure 11a), as suggested in previous works for graphene [86], linoleic acid [87] and OA-coated CaO NPs [47], would lead to a more polar/hydrophilic film surface. However, other possible interactions might exist between the Ga/PLA nanocomposite components (Figure 11a). PLA could also interact with the Ga-oxide-hydroxide surface layer of Ga/OA NPs through hydrogen bonding (dashed lines in Figure 11a), as was proposed by several researchers for SiO_2_ [61,88], TiO_2_, ZnO [74] and Fe_3_O_4_ [89] in composites with PLA. This would cause exposure of the non-polar parts of PLA chains at the surface, leading to higher hydrophobicity of the films. Indeed, our FTIR results indicate the possible involvement of carbonyl groups (slightly increased fraction of bound carbonyls), while OH vibrations are too weak for any conclusions.

The strength of the interactions between the Ga/OA NPs and PLA can also be indirectly evaluated through the observed thermal properties. The change in the heat capacity at glass transition is decreased for Ga/PLA with regard to PLA, especially in the case of Ga/PLA0.4%, which also exhibited the highest Young’s modulus. Both characteristics can be explained as a result of the increased formation of a rigid (immobile) amorphous fraction of PLA around the finely dispersed Ga/OA NPs due to interfacial interactions [61,73,88,90]. Among the three compared nanocomposites, Ga/PLA0.4% contained the largest fraction of amorphous PLA (lowest crystallinity) and the smallest mobile amorphous fraction (the lowest Δ*C_p_* at *T*_g_); therefore, its rigid amorphous fraction was the largest [61,74,88,90]. This means that PLA chains are less mobile, so a larger force (stress) needs to be applied for a chosen displacement (strain). It is also somewhat reflected in the *T*_g_, which, although only slightly increased, is the highest for Ga/PLA0.4%.

Such effect of nanofillers on the PLA due to interfacial interactions is quite common [40,61,73,88,90,91]. However, if these interactions are quite strong, such as in hydrogen bonding, they usually also cause a suppression of nucleation [73,74]. Conversely, in the case of Ga/PLA, we observed increased nucleation (decreased *T*_cc_) with increasing Ga content in the nanocomposites. This implies that PLA–OA van der Waals interactions predominate over Ga-PLA hydrogen bonding within the Ga/PLA nanocomposites. Thus, the presence of nanogallium loosens PLA molecular packing and increases free volume rather than reducing it by constraints in chain segmental mobility [88]. A decrease in both Δ*C_p_* and *T*_cc_ has also been observed in PLA composites with silica, graphene, graphene oxide, carbon nanotubes, WS_2_ nanotubes, and Ag NPs [61,73,88,91,92]. It is also possible that the nucleating effect of Ga/OA NPs (additional sites for crystallization at the interfaces) is so strong that it overcomes the effect of segmental chain mobility retardation at the interface with NPs. In fact, the nucleating effect of Ga/OA NPs is quite strong, as it led to semicrystalline PLA in Ga/PLA0.8% during solvent casting at room temperature. With higher nanogallium loadings, agglomeration comes into play, which weakens the interfacial immobilization and nucleation as aggregates become too large to embed into the growing PLA crystals [90]. This results in lower crystallinity and Young’s modulus (E) at a higher content. Maximum E in relation to filler content has been observed before, e.g., in PLA composites with SiO_2_/OA NPs [93], TiO_2_ NPs [94], or WS_2_ nanotubes [91].

Hence, while we admit the possibility of stronger hydrogen bond interactions between Ga/OA and PLA, we conclude that weaker van der Waals interactions prevail in the Ga/PLA nanocomposites prepared with solvent casting from chloroform, which leads us back to the question of increased hydrophobicity. It can be explained if a fraction of OA is present at the Ga/PLA surface, either as a free molecule (the FTIR spectrum of Ga/OA NPs points to a possible excess OA) or within Ga/OA NPs at the surfaces that are not fully covered with PLA. Previous investigations have discovered a fraction of OA-coated NPs at the surface in solvent-cast nanocomposite films [95]. Further and more detailed analyses will be needed to draw firmer conclusions about this topic.

Our results demonstrate that the addition of nanogallium provided the PLA with antibacterial protection against PAO1 bacteria at the surface. From the 0.8 wt.% Ga content onwards, the growth of bacteria in contact with the film was inhibited in a nutrient broth, while their cross-feeding through intercellular nanotubes in the absence of nutrients was prevented. The non-remote but rather contact-based antibacterial action of Ga/PLA could be understood as a consequence of the unwrapping and exposure of a fraction of Ga/OA NPs or perhaps also Ga/OA/PLA NPs, which broke off from the nanocomposite film at/close to the surface but did not diffuse away from the film into the medium due to their hydrophobicity (Figure 11b). Thus, a high gradient is established, and Ga concentration is much higher at the surface of the Ga/PLA for bacteria in the vicinity or in contact with the film. In addition, there could be a contribution of generally slower bacterial growth on films than planktonically in a liquid medium [96].

As PLA degrades slowly, such an evident release of Ga/OA NPs from the Ga/PLA already occurring within 24 h at 37 °C is somewhat unexpected and surprising. We propose two possible explanations for it, which are not necessarily exclusive:(a)Ga/OA NPs enhance the degradation of PLA. This is expected as a result of the higher thermal conductivity of the metallic liquid Ga core [27], leading to a higher temperature in the vicinity of Ga/OA NPs and catalytic activity of the gallium oxide/hydroxide surface layer, which could accelerate the hydrolytic degradation of PLA as with ZnO, CaO or silicate nanofillers [47,97,98], with the additional possible enhancement by a local change in pH due to the dissolution of Ga, as in the case of Mg [99].(b)Weak interactions between Ga/OA NPs and PLA enable the easy separation of these two nanocomposite components, especially when additional stronger interactions with water and other liquid broth components are possible. Water helps in this polymer chain rearrangement with its plasticizing effect on PLA (it enhances segmental mobility) [100], especially at a raised temperature (37 °C), where the above-mentioned higher thermal conductivity of metallic (Ga) nanofiller also plays an important role in exposing the nanoparticles from the Ga/PLA nanocomposite. In addition, we need to consider a possible fraction of uncovered Ga/OA NPs at the surface, for which only the neighboring PLA chains would need to move away to eject the NPs.

Once Ga NPs are exposed at the surface of the film, they affect the bacteria in contact. Moreover, bacteria can trigger the release of Ga NPs from Ga/PLA by enhancing the degradation of PLA. Namely, *P. aeruginosa* can cause the severe degradation of PLA at around 30 °C with its esterase enzyme [101], which is one of its secreted virulence factors [102]. Thus, the Ga/PLA material could respond in a way that leads to stronger virulence with faster degradation and the enhanced release of antimicrobial NPs.

While increased hydrophobicity of Ga/PLA with respect to PLA could be beneficial in certain applications, it can also cause low cell affinity and induce an inflammatory response in the body [2]. This is already an issue with pure PLA. Here, we solved it by adding poly(L-lysine) onto the surface of the films, successfully adhering to and growing human HaCaT keratinocytes on all Ga/PLA films. This sounds promising even for more internal biomedical applications of Ga/PLA nanocomposites, but many additional in vitro studies on different cell lines will be needed for a more accurate evaluation of its potential. Moreover, it would be important to examine whether Ga/PLA films can inhibit bacteria and enable the growth of HaCaT cells on them at the same time, under the same conditions, as it might be possible that the polylysine coating reduces or enhances contact antibacterial activity, or that a fraction of Ga NPs are removed from the surface during exchange with a fresh medium, which makes the surface more hospitable for the spreading of keratinocytes. Indeed, for Ga/PLA2.4%, we observed an inhibition of growth during the first day, which did not continue on the second day. However, as we noticed, polylysine could also be washed away, which would lead to a less attractive surface for HaCaT cells, and it would be worth exploring how prolonged this antibacterial action can be. Furthermore, the analyzed materials in this research were cut pieces of solvent-cast films. It would be very interesting and meaningful to see how this material performs in other shapes, such as drawn films, coatings, or fibers. We have very recently shown how uniaxially drawn PLA-based nanocomposites can direct and polarize HaCaT keratinocytes through the ultrasonically triggered piezoelectric effect [103], while another recent investigation added 300 nm Ga particles onto different fibers (cellulose-polyester, cotton, nylon, and linen) and converted them to Ga/Cu particles with strong antimicrobial and antiviral properties, with the Ga part enabling good attachment to the fibers [104]. These are a few possible further directions in the investigation and development of Ga/PLA.

## 5. Conclusions

The solvent casting of colloidal Ga/OA NPs in a chloroformic solution of PLA yielded a homogeneous nanocomposite material, which exhibited a larger water wetting angle, Young’s modulus, ordering of the PLA phase, and rigid amorphous fraction due to predominantly van der Waals interactions between Ga/OA NPs and PLA. A fraction of Ga NPs evidently had already emerged from the nanocomposite film within 24 h and affected *P. aeruginosa* PAO1 bacteria at or close to its surface. Nanogallium/PLA composite is a new nanocomposite material, and, to our knowledge, its contact antibacterial properties have been shown for the first time in this investigation. Good biocompatibility with human keratinocytes forecasts its various potential biomedical applications. Its further development in this direction could be tailored through the intriguing influences of the non-crystalline Ga/OA NPs on the structural ordering and degradation of the surrounding PLA, which calls for more intensive investigation.

## Figures and Tables

**Figure 1 pharmaceutics-16-00228-f001:**
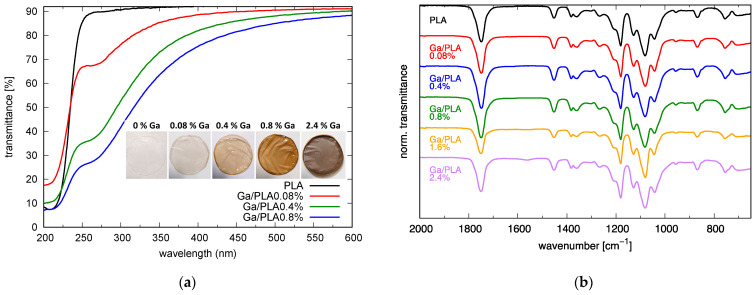
Ga/PLA nanocomposites: (**a**) Appearance and optical properties; (**b**) ATR FTIR spectra.

**Figure 2 pharmaceutics-16-00228-f002:**
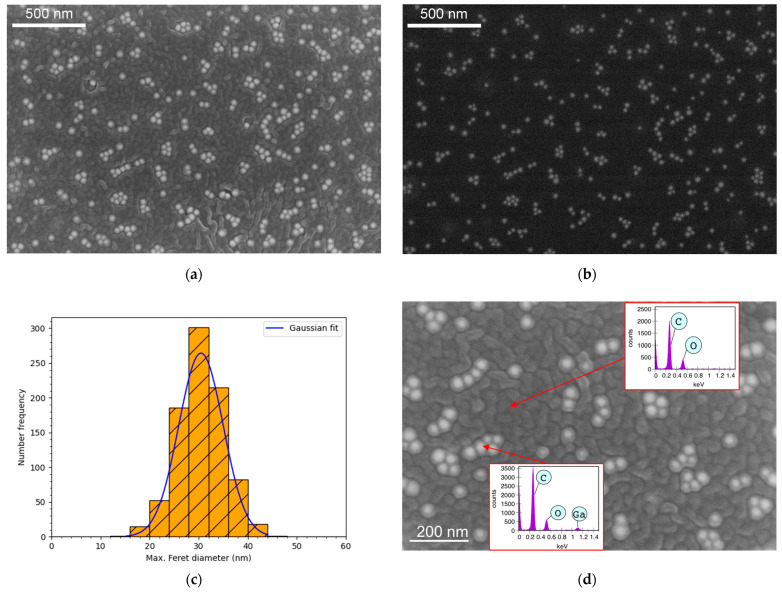
Detailed morphology of Ga/PLA0.8%: (**a**) SEM image from secondary electrons; (**b**) SEM image from backscattered electrons; (**c**) size distribution of Ga/OA NPs in the Ga/PLA nanocomposite based on backscattered SEM images; (**d**) EDXS analysis with one spectrum taken on a few bright nanospheres and the other on the darker wrinkled material (PLA) in between.

**Figure 3 pharmaceutics-16-00228-f003:**
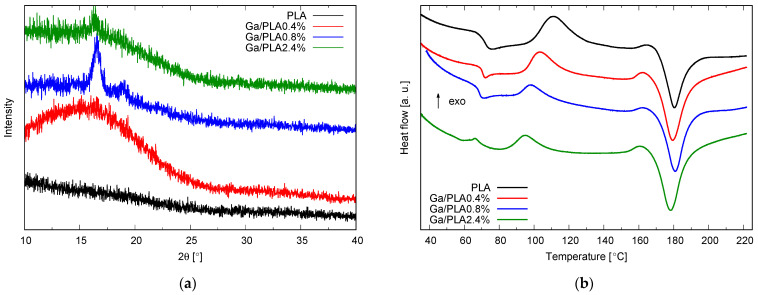
Crystallinity of films: (**a**) X-ray diffractograms of PLA and Ga/PLA with different Ga contents; (**b**) DSC curves of the same PLA and Ga/PLA.

**Figure 4 pharmaceutics-16-00228-f004:**
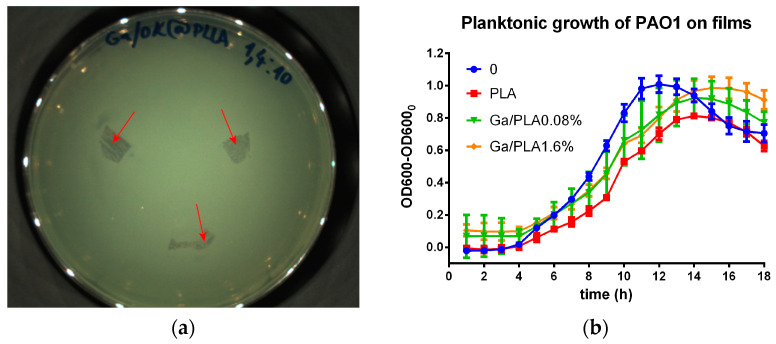
Antibacterial tests on PAO1: (**a**) Diffusion antibiogram for Ga/PLA1.6% (three pieces of this film are marked by red arrows); (**b**) planktonic growth in the presence of PLA and Ga/PLA in MH broth.

**Figure 5 pharmaceutics-16-00228-f005:**
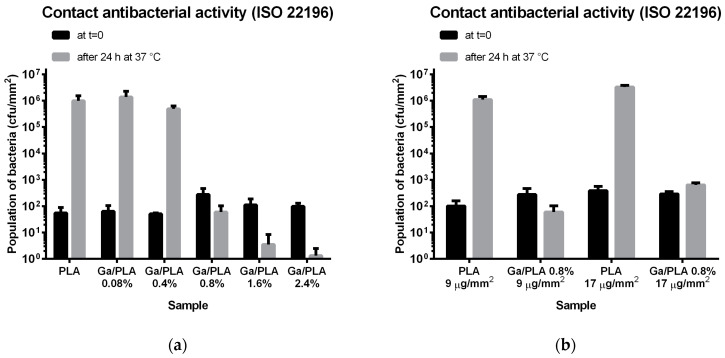
Contact antibacterial action in MH medium according to ISO 22196 standard test [67]: (**a**) Population of bacteria before and after 24-h incubation in contact with nanocomposite films with different Ga content; (**b**) comparison of two films with the same Ga content but different overall masses/thicknesses; (**c**) SEM morphology of the PAO1 bacteria on the PLA film after 24-h incubation; (**d**) SEM morphology of the PAO1 bacteria on the Ga/PLA1.6% film after 24-h incubation.

**Figure 6 pharmaceutics-16-00228-f006:**
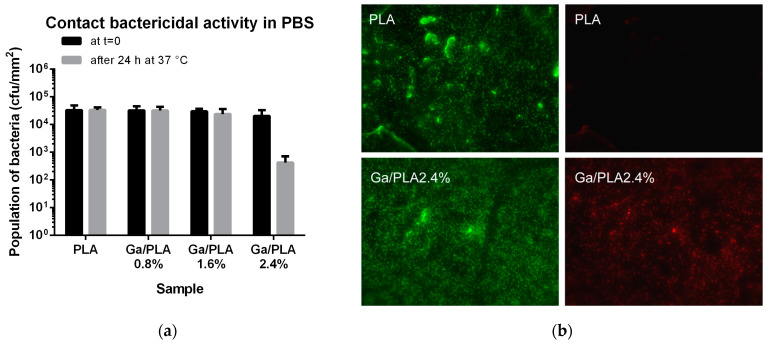
Contact bactericidal activity in PBS: (**a**) Population of PLA and Ga/PLA films with bacteria at 0 and 24 h of incubation; (**b**) live/dead staining of bacteria on the surface of PLA (**top**) and Ga/PLA2.4% (**bottom**). All bacteria (live and dead) are colored green, while dead bacteria additionally emit red fluorescence.

**Figure 7 pharmaceutics-16-00228-f007:**
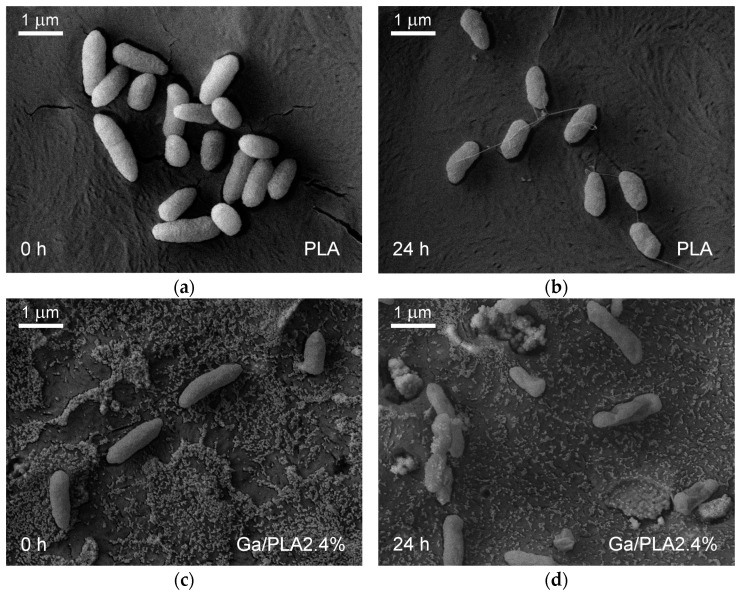
SEM images of bacteria from the PBS on PLA or Ga/PLA before (**a**,**c**) and after 24 h of incubation at 37 °C in contact with the films (**b**,**d**).

**Figure 8 pharmaceutics-16-00228-f008:**
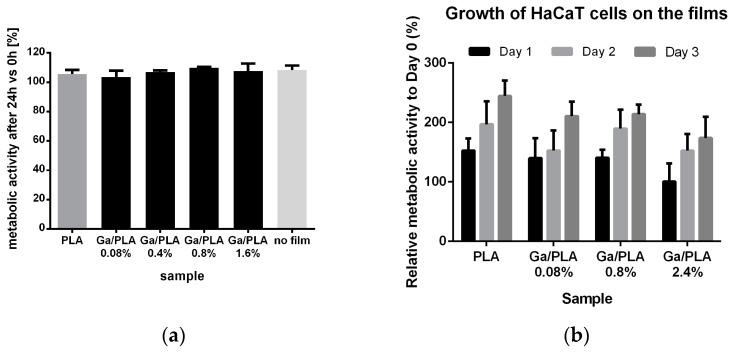
Cytocompatibility of Ga/PLA nanocomposites: (**a**) Metabolic activity of HaCaT keratinocytes on the tissue-culture-treated surface in the presence of PLA and Ga/PLA films; (**b**) growth of HaCaT keratinocytes on polylysine-coated PLA and Ga/PLA films.

**Figure 9 pharmaceutics-16-00228-f009:**
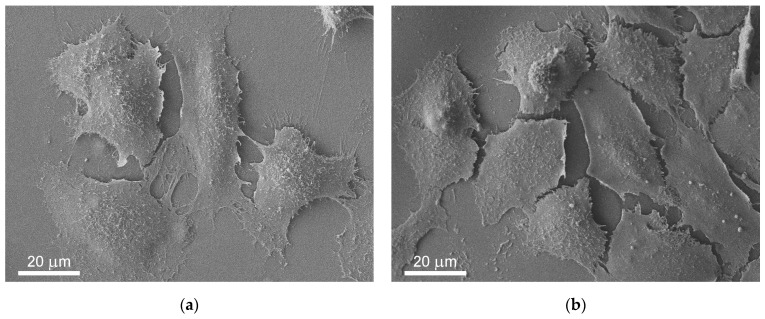
SEM images of HaCaT keratinocytes after three days of growth on the following films: (**a**) PLA; (**b**) Ga/PLA0.08%; (**c**) Ga/PLA0.8%; (**d**) Ga/PLA2.4%.

**Figure 10 pharmaceutics-16-00228-f010:**
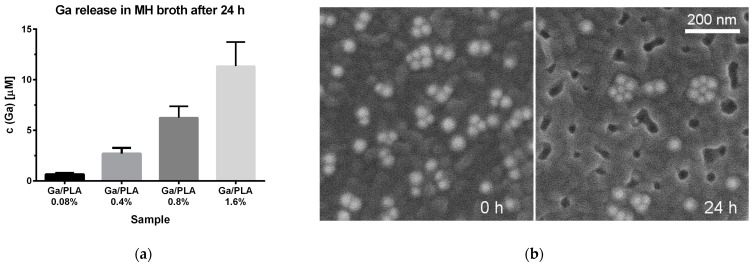
(**a**) Released Ga from Ga/PLA films under the conditions of a planktonic antibacterial test but in absence of bacteria; (**b**) surface of Ga/PLA0.8% film before (**left**) and after (**right**) 24 h of soaking and orbital shaking in MH broth at 37 °C.

**Figure 11 pharmaceutics-16-00228-f011:**
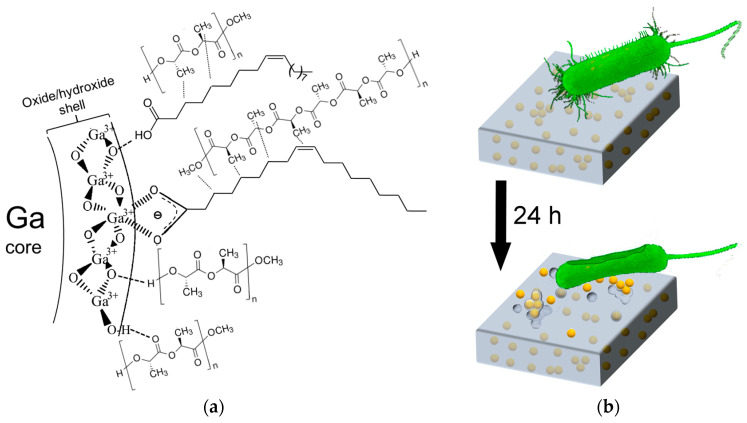
Ga/PLA nanocomposite: (**a**) schematic of possible interactions within the nanocomposite; (**b**) contact antibacterial action.

**Table 1 pharmaceutics-16-00228-t001:** Thermal, wetting, and mechanical properties of PLA and Ga/PLA films.

	PLA	Ga/PLA0.4%	Ga/PLA0.8%	Ga/PLA2.4%
*T*_g_ [°C]	69.1	70.0	68.0	70.4
Δ*C_p_*_,g_ [J/(gK)]	1.00	0.66	0.87	0.77
*T*_cc_ [°C]	110.9	103.4	98.3	94.8
Δ*H*_cc_ [J/g]	32.7	22.9	10.1	15.4
*T*_α’→α_ [°C]	165.6	163.1	163.1	162.0
Δ*H*_α’→α_ [J/g]	2.4	8.1	4.5	12.6
*T*_m_ [°C]	180.5	179.4	181.0	178.3
Δ*H*_m_ [J/g]	45.6	38.0	49.6	35.9
*X*_c, DSC_ [%]	12.4	14.7	39.2	20.4
*X*_c, FTIR_ [%]	15.0 ± 0.7	18 ± 4	30 ± 1 **	20 ± 2
*θ*_H2O_ [°]	70 ± 4	75 ± 3 **	83 ± 6 **	89 ± 7 **
E [GPa]	1.6 ± 0.3	3.9 ± 0.4 *	2.5 ± 0.6	2.4 ± 0.6

* statistically significant vs. PLA (*p* < 0.05); ** statistically significant vs. PLA (*p* < 0.005).

## Data Availability

All data generated or analyzed during this study are included in this published article and its supplementary information files. The source files are available in the corresponding author’s storage and will be shared upon request.

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
