# Peer review of "Nanogallium-poly(L-lactide) Composites with Contact Antibacterial Action"

_pharmaceutics, 2024, doi:10.3390/pharmaceutics16020228_

Round 1

Reviewer 1 Report

Comments and Suggestions for Authors

·       Keywords must be more effective and also technical.

·       About the Introduction section to make the reading clearer and smoother it should be organized according to the following items: i) present state of the art; ii) literature review; iii) motivation and objective of the study proposed; iv) innovative contribution in terms of methodology developed.

·       Check the grammar throughout the manuscript.

·       Authors must correct some typological errors present in the manuscript and also put section number because it is easy to review the manuscript.

·       Authors read guide for authors and then align the manuscript carefully according to the Journal format.

·       In the sec 2.1-line no.130, authors should rewrite it.

·       In the sec 2.2-line no. 142 why authors should take “Ga solution was used to prepare a 0 % Ga reference material”, explain in detail.

·       Add some new references like 2021, 2022 and 2023 in the manuscript.

Comments on the Quality of English Language

·       Keywords must be more effective and also technical.

·       About the Introduction section to make the reading clearer and smoother it should be organized according to the following items: i) present state of the art; ii) literature review; iii) motivation and objective of the study proposed; iv) innovative contribution in terms of methodology developed.

·       Check the grammar throughout the manuscript.

·       Authors must correct some typological errors present in the manuscript and also put section number because it is easy to review the manuscript.

·       Authors read guide for authors and then align the manuscript carefully according to the Journal format.

·       In the sec 2.1-line no.130, authors should rewrite it.

·       In the sec 2.2-line no. 142 why authors should take “Ga solution was used to prepare a 0 % Ga reference material”, explain in detail.

·       Add some new references like 2021, 2022 and 2023 in the manuscript.

Author Response

We thank the reviewer for reading our manuscript and giving us useful suggestions for its improvement. Below are our replies to reviewer's comments and questions:

  1. Keywords must be more effective and also technical.

We have modified the keywords to make them more specific and effective.

  1. About the Introduction section to make the reading clearer and smoother it should be organized according to the following items: i) present state of the art; ii) literature review; iii) motivation and objective of the study proposed; iv) innovative contribution in terms of methodology developed.

The introduction is organised in the following way: i) present state of the art (properties and today’s applicability of PLA) in lines 29-56; ii) research problem establishment (bacterial contamination of PLA) with literature overview on PLA composites with antimicrobials in lines 57-68; iii) innovative contribution/solution and motivations with literature support of the hypotheses (choice of Ga NPs as filler for composite) in lines 69-104; and iv) objective/aim in the end (lines 105-109), according to the journal guidelines. This paper is not about new methodology but about the development and investigation of a new composite material.

  1. Check the grammar throughout the manuscript.

Thank you for this suggestion. We have found a few grammar errors and corrected them after one more detailed reading of the manuscript. Then, we have also submitted our manuscript for English language editing service that was provided by MDPI. Thus, the revised manuscript has been proofread and corrected by a native English speaker. All the corrections are highlighted in yellow in the revised version.

  1. Authors must correct some typological errors present in the manuscript and also put section number because it is easy to review the manuscript.

Thank you. We have corrected typographical errors in lines 10, 15, 35, 103, 227, 261, 271, 315, 324, 325, 335, 344, 348, 402, 413, 465, 472, 565, 620, 668 (see the modified version of the manuscript, in which they are yellow highlighted). There are already section numbers in the first version of the manuscript. In fact, the reviewer is referring to them in comments no. 6 and 7.

  1. Authors read guide for authors and then align the manuscript carefully according to the Journal format.

We have corrected the text alignment in lines 546-557 (now lines 560-571) from left-aligned to justified on both sides. We have also checked the formatting of the reference citations in the bibliography at the end and corrected any encountered mistakes (highlighted in yellow). Otherwise, we have considered the instructions for authors already in the first version of the manuscript. We also used the provided template of the Pharmaceutics journal and carefully followed the instructions in it.

  1. In the sec 2.1-line no.130, authors should rewrite it.

Thank you. We have divided the long sentence that extended to line 130 into two shorter sentences (lines 133-135 in the revised version of the manuscript, also highlighted in yellow): “The supernatant (pale yellow) was discarded again and the sediment was redispersed in OA and chloroform. After the addition of ethanol, the suspension was centrifuged at 8000×g for 20 minutes.”

  1. In the sec 2.2-line no. 142 why authors should take “Ga solution was used to prepare a 0 % Ga reference material”, explain in detail.

The full sentence is: “PLA solution undergoing the same procedure but without the addition of any colloidal Ga solution was used to prepare a 0 % Ga reference material.” We added “(PLA film)” at the end of this sentence in the revised manuscript to make it clear what the 0 % Ga reference material was. The purpose of PLA without Ga (or 0 % Ga reference) was to serve as a blind or negative reference that should be as similar as possible to the investigated samples. Therefore, we used the same procedure as for Ga/PLA to prepare pure PLA films without any Ga NPs, so that the effect of Ga NPs addition was evident.

  1. Add some new references like 2021, 2022 and 2023 in the manuscript.

There are already 11 references from 2021, 11 references from 2022 and 4 references from 2023 in the first version of the manuscript. We added one more reference from 2022 at a place where a reference was missing (reference no. 16 in line 53).

Reviewer 2 Report

Comments and Suggestions for Authors

The publication by Mario Kurtjak et al. entitled

Nanogallium-poly(L-lactide) composites with contact antibacterial action is in a scope of journal Pharmaceutics MDPI and recommended publication.

The manuscript reports on synthesis of Ga-poly(L-lactide) (PLA) composites of various content of Ga (from 0.08 wt.% to 2.4 wt.%) for bactericidal application against Pseudomonas aeruginosa bacteria.

The prepared composites morphology, crystallinity, wettability, optical, thermal and mechanical properties were studied by SEM, XRD, FTIR, spectrophotometric method, DSC, contact angle and dynamic angle analyzers.

The antibacterial and cytotoxic properties to Pseudomonas aeruginosa bacteria and human HaCaT keratinocytes, respectively, were studied by spectrophotometry, contact angle and SEM.

The aim of the research is well justified.

Introduction part provides a satisfactory number of references and refers to a broad range of literature on PLA metal (oxide) NPs applied as bactericidal agent. Application of Ga NPs as bactericidal and antitumor agent is also well justified. The objective of the research to prepare homogeneous composites of Ga NPs and PLA, study the influence of Ga and its quantity in PLA on composite properties, and the antibacterial activity against Pseudomonas aeruginosa is satisfied. As it was pointed by the authors Ga NPs incorparated in the other polymers than PLA has been already investigated, although Ga-PLA composite was studies for the first time.

The obtained properties of the prepared Ga-PLA composite are convincing for further biomedical applications.

The manuscript describes and refers to interesting methodology of characterization of several parameters such as evaluation of crystallinity from DSC curved and FTIR, and spectrophotometric quantitative evaluation of Ga content. It is a valuble contribution in the research of NPs-polymer specimen for biomedical application.

Only few misprints were found, e.g.

  • Line 400 [Pan2007, Pan2008, Chen2011]

Checking misprints is recommended.

Author Response

We thank this reviewer for recognising the importance and quality of our work and recommending it for publication.

Also thank you for noticing and warning about misprints. The one in line 400 (now line 413) has been corrected. After reading through the manuscript carefully once again, we discovered and corrected several other misprints in lines 10, 15, 35, 103, 227, 261, 271, 315, 324, 325, 335, 344, 348, 402, 413, 465, 472, 565, 620, 668 (see the modified version of the manuscript, in which the corrections are highlighted in yellow). There was also wrong text alignment in lines 546-557 (now lines 560-571) and a few errors in the bibliography citations. We have also submitted our paper for additional proofreading by the MDPI English editing service (all the corrections are highlighted in yellow in the revised version of the manuscript).

Reviewer 3 Report

Comments and Suggestions for Authors

Authors developed homogenous composite films from Ga NPs and PLA and do lots of characterization and in vitro studies. Even with many of findings, there are some issues need to address before accepting.

1.     The composite is hydrophobic, the solvent that used in this manuscript was chloroform. How to conduct the cytotoxicity of hydrophobic nanoparticles by using the cells? Did authors resuspended in medium for treating cells, how to avoid the sediment when treating cells or some in vivo future study? What’s the future use of those hydrophobic nanoparticles?

2.     In the abstract part, authors named this composite as a promising new biopolymer-based materials. Please correct it to polymer-based materials, not bio.   Too many keywords. Normally it should be 3-5 keywords.

3.     Line 50. “Substituting the hardly degradable plastics in the masks with PLA could solve this issue”? Please cite reference. Line 81-82, please also cite the Ref.

4.     What the molecule weight of PLA that used, why? PLGA has been widely used than PLA for nanoparticles preparation, what about using PLGA?

5.     Line 141, how to keep the total mass/surface area constant? Did authors measure the surface area? How?

6.     There would be a stronger hydrogen bond interaction between Ga/OA and PLA, however, authors concluded that van der Waals interactions as predominantly one, why?

7.     Generally, bacteria grow much slower on films than planktonically in a liquid, how to put this into account?

Comments on the Quality of English Language

English language need extensive editing. 

Author Response

We thank the reviewer for reading our manuscript and giving us useful suggestions for its improvement. The revised manuscript has an improved level of English language. First, we have gone through the text once more and corrected several misprints and also a few encountered grammar mistakes. Then, we submitted the text for English editing service provided by MDPI. Thus, our manuscript has been proofread and corrected by a native English speaker now. All the corrections are highlighted in yellow in the revised version of the manuscript. Below are our replies to reviewer's comments and questions:
  1. The composite is hydrophobic, the solvent that used in this manuscript was chloroform. How to conduct the cytotoxicity of hydrophobic nanoparticles by using the cells? Did authors resuspended in medium for treating cells, how to avoid the sediment when treating cells or some in vivo future study? What’s the future use of those hydrophobic nanoparticles?

The composite was in the form of films not nanoparticles. Therefore, it could only be inserted into a medium, not resuspended. We already addressed the hydrophobicity problem and possible future directions in the first version of the manuscript, in the last part of Discussion. Hydrophobicity is an issue with biomedical application of PLA in general and there are different approaches to solving it. In this investigation we used polylysine coating and we observed good cytocompatibility, attachment and growth of HaCaT keratinocytes directly on the surface of initially hydrophobic Ga/PLA films. However, we are aware that other complications might emerge due to agglomeration of released Ga/OA NPs in future hypothetical applications inside the body, although there are possible options for their conversion to hydrophilic nanoparticles in biological milieu, further dissolution to Ga(III) ions and elimination from the body. On the other hand, there are many other possible biomedical applications that do not involve insertion in vivo yet require antibacterial properties. Several are mentioned in the Introduction and in the last part of the Discussion. For such applications, these issues will not be of concern.

  1. In the abstract part, authors named this composite as a promising new biopolymer-based materials. Please correct it to polymer-based materials, not bio.   Too many keywords. Normally it should be 3-5 keywords.

We have removed the »bio« prefix. We have reduced the number of keywords from 8 to 6. According to the journal guidelines/instructions for authors: “Three to ten pertinent keywords need to be added after the abstract.”

  1. Line 50. “Substituting the hardly degradable plastics in the masks with PLA could solve this issue”? Please cite reference. Line 81-82, please also cite the Ref.

Thank you for noticing the missing references. One reference (new reference no. 16) added in line 50 (53 in the revised version of the manuscript):

Soo, X.Y.D.; Wang, S.; Yeo, C.C.J.; Li, J.; Ni, X.P.; Jiang, L.; Xue, K.; Li, Z.; Fei, X.; Zhu, Q.; et al. Polylactic acid face masks: Are these the sustainable solutions in times of COVID-19 pandemic? Sci. Total Environ.2022, 807, 151084, doi:10.1016/j.scitotenv.2021.151084.

And one (previous reference no. 48, now no. 49) in line 83 (87 in the revised version of the manuscript), at the end of this sentence, which begins in line 81 (85 in the revised version): “There was no significant increase in generation of reactive oxygen species (ROS) for L929 mouse fibroblasts in the presence of nanogallium-hydroxyapatite composite compared to hydroxyapatite alone.”

  1. a) What the molecule weight of PLA that used, why?

We used commercial poly(L-lactide) Resomer L207S purchased from Evonik: https://healthcare.evonik.com/en/medical-devices/bioresorbable-polymers/standard-polymers

Using the available data from the provider (1.5-2.0 dl/g inherent viscosity), its molecular weight can be estimated as 100000-140000 Da, based on Mark-Houwink equation: ? = K*M?, with K= 22.1·10-5 dl/g, α = 0.77 for PLA in CHCl3 (parameters from R. Köhn, Z. Pan, J. Sun, C. Liang, Catalysis Communications: 2003, Vol. 4, Issue 1, 33-37).

Poly(L-lactide) with high molecular weight is expected to keep the antibacterial nanoparticles embedded and retain them at the surface for a longer time due to its slower degradation, which is favourable for contact-based antimicrobial surfaces. High-molecular-weight PLA is also more suitable for designing medical devices, such as load-bearing implants, orthopaedic fixation devices or external devices and protective equipment, which need to maintain their mechanical properties over an extended period. Moreover, the effect of nanofiller on the structural ordering of the PLA chains within the composite is more easily detectable and more meaningful in such PLA, as it can lead to crystallisation, which also affects other properties, e.g., thermal, mechanical or piezoelectric properties.

b) PLGA has been widely used than PLA for nanoparticles preparation, what about using PLGA?

Yes, PLGA has favourable properties for the preparation of polymer nanoparticles and drug delivery inside the body if faster degradation is desired. On the other hand, PLA, with a longer degradation time, is more suitable for designing medical devices and functional surfaces; it can also create piezoelectricity. We believe that PLGA could be used in a very similar way as PLA to form a nanogallium/PLGA composite since PLGA is also soluble in chloroform. It would definitely be interesting to compare the two polymer matrices in the future. However, in this research, we did not use PLA for nanoparticle preparation but for the preparation of films with dimensions of several centimetres and thickness of 15-30 ?m, in which Ga NPs were embedded. We aimed to design contact-based antimicrobial surfaces, for which PLA with slower degradation and better mechanical properties is preferred over PLGA.

  1. Line 141, how to keep the total mass/surface area constant? Did authors measure the surface area? How?

Yes, we used a ruler to measure the diameter of the glass Petri dish and the film. The samples were in the form of films or disc-shaped sheets, if that's more understandable. The smooth films can be easily weighed after being peeled off from the Petri dish. Here's also an example of our calculation of mass/surface area.

Solution of PLA+Ga/OA NPs in chloroform containing a total mass of 40 mg of composite components is poured onto a glass Petri dish with internal diameter 55 mm and dried to form a Ga/PLA film with the same diameter. Hence, mass/surface area is 40 mg/(?*552 mm2/4) = 17 µg/mm2. Now, we upscale this to a Petri dish with internal diameter of 77 mm to make a bigger film. How much composite components do we add to keep the same mass/surface area? m = 17 µg/mm2 * ?*772 mm2/4 ≈ 80 mg.

  1. There would be a stronger hydrogen bond interaction between Ga/OA and PLA, however, authors concluded that van der Waals interactions as predominantly one, why?

As explained in the Discussion section of the manuscript, the signal of bound carbonyls in FTIR was not strong enough to support predominantly hydrogen bonding with carbonyl groups of PLA, while hydrogen bonding through OH groups of PLA could not be detected due to weak signal of these functional groups in FTIR ATR. Moreover, predominantly strong interaction, such as hydrogen bonding, would lead to suppressed nucleation, i.e., delayed cold crystallisation (as observed in several recent investigations of PLA composites with different fillers, see references in the manuscript), while we observed the opposite - PLA chains around Ga/OA NPs are mobile enough to crystallise at lower temperature than in pure PLA. Furthermore, hypothesis of predominantly van der Waals interactions explains better the observed fast escape of Ga/OA NPs from the composite at the surface within 24 hours.

We hypothesize that hydrogen bonding between Ga/OA and PLA is not frequently established during this solvent casting procedure due to strongly bound OA on Ga NPs, which occupies the nanoparticle surface and creates steric hindrances for PLA chains. OA can also prevent PLA chains from accessing the Ga NP surface through interactions with them.

  1. Generally, bacteria grow much slower on films than planktonically in a liquid, how to put this into account?

Yes, we considered this fact as one possible reason for the observed contact-based antibacterial effect without any effect on planktonic bacteria. However, they could grow normally within 24 hours on the PLA reference and composites with low Ga contents. Therefore, slower growth on films is less important factor here, so we rephrased that sentence in the new version of the manuscript as it seemed to have overemphasized it.

Round 2

Reviewer 1 Report

Comments and Suggestions for Authors

Authors corrected the reviewer comments clearly, So the manuscript has been accepted.

Reviewer 3 Report

Comments and Suggestions for Authors

Can be accepted with current version.